# The Phenomenon of Clopidogrel High On-Treatment Platelet Reactivity in Ischemic Stroke Subjects: A Comprehensive Review

**DOI:** 10.3390/ijms21176408

**Published:** 2020-09-03

**Authors:** Adam Wiśniewski, Karolina Filipska

**Affiliations:** 1Department of Neurology, Collegium Medicum in Bydgoszcz, Nicolaus Copernicus University in Toruń, Skłodowskiej 9 Street, 85-094 Bydgoszcz, Poland; 2Department of Neurological and Neurosurgical Nursing, Collegium Medicum in Bydgoszcz, Nicolaus Copernicus University in Toruń, Łukasiewicza 1 Street, 85-821 Bydgoszcz, Poland; karolinafilipskakf@gmail.com

**Keywords:** platelets, ischemic stroke, clopidogrel, resistance, platelet reactivity, antiplatelet therapy, platelet function

## Abstract

Clopidogrel is increasingly being used for the secondary prevention of ischemic stroke according to the updated guidelines on acute stroke management. Failure to achieve a drug response is referred to as clopidogrel resistance. Similarly, a higher activation of platelets during clopidogrel therapy—high on-treatment platelet reactivity—is equivalent to a reduced effectiveness of a therapy. Clopidogrel resistance is considered to be a common and multifactorial phenomenon that significantly limits the efficacy of antiplatelet agents. The aim of the current study is to review the latest literature data to identify the prevalance and predictors of clopidogrel high on-treatment platelet reactivity among stroke subjects and to establish the potential impact on clinical outcomes and prognosis. Clinical databases were searched by two independent researchers to select relevant papers on the topic, including all types of articles. Several important predictors contributing to clopidogrel resistance were identified, including genetic polymorphisms, the concomitant use of other drugs, or vascular risk factors, in particular nonsmoking and diabetes. Clopidogrel high on-treatment platelet reactivity has a negative impact on the clinical course of stroke, worsens the early- and long-term prognoses, and increases the risk of recurrent vascular events. Platelet function testing should be considered in selected stroke individuals, especially those predisposed to clopidogrel resistance, for whom an improvement in the efficacy of antiplatelet therapy is essential. This particular group may become the greatest beneficiaries of the modification of existing therapy based on platelet function monitoring.

## 1. Introduction

Clopidogrel is one of the most widely used antiplatelet agents worldwide. It has an established position as an antagonist of platelet function, leading to a reduced risk of recurrent ischemic events. It plays a primary role in the prevention of coronary heart disease in the field of cardiology. Due to the updated guidelines on stroke, this drug is also increasingly being used for the secondary prevention of ischemic stroke [1]. It might be used as a monotherapy as a a reasonable and equivalent option to aspirin in nonembolic ischemic stroke patients or in combination with aspirin for 21 days of dual antiplatelet therapy following a minor ischemic stroke or transient ischemic attack (TIA) in patients with a higher risk of recurrent stroke. Some relevant papers have demonstrated its efficacy and safety and, in some cases, its advantage over aspirin [2]. However, the inhibition of platelets among different subjects might be limited and variable. This phenomenon is caused by multiple factors, leading to a reduced response of platelets, which may limit the therapeutic effect of the drug. Failure to achieve a drug response is referred to as clopidogrel resistance or high on-treatment platelet reactivity, and this has become the most important factor inhibiting the antiplatelet effect of the agent, resulting in ineffectiveness [3]. Therefore, various assays are used to assess platelet function for monitoring antiplatelet therapy [4,5]. They not only allow the phenomenon to be detected but can also estimate the level of platelet inhibition, thus allowing the effectiveness of the therapy to be assessed. It is extremely important for stroke subjects for whom responsiveness to antiplatetet agents is crucial.

Unfortunately, knowledge of the role of clopidogrel resistance in ischemic stroke is not well established. In contrast, there are numerous reports on coronary heart disease that encourage individual or personalized treatments based on platelet function testing [6,7]. There is still a lack of large, multicenter, randomized clinical trials regarding ischemic stroke. Nevertheless, the relevant body of literature contains some interesting cross-sectional, cohort, and observational studies that deal with this subject. The aim of the current study is to review the latest literature data to identify the prevalance of clopidogrel high on-treatment platelet reactivity among stroke subjects—its mechanisms, risk factors, and potential impact on the prognosis and clinical outcome, and its association with recurrent vascular events.

## 2. Molecular Mechanism of Action of Clopidogrel

Clopidogrel is a pro-drug and is inactive in vitro. It must be biotransformed in the human body after oral ingestion to become operational. Antiaggregative properties are produced by an active shortlisting metabolite generated by the cytochrome P450 (CYP)-dependent pathway in the liver. This is contradictory to other antiplatelet agents, such as aspirin or ticagrerol, which are active from the beginning and do not need to undergo transformation. The inhibition of platelets is a result of the irreversible binding of P2Y12 receptors. Adenosine diphosphate (ADP) activates platelets by binding the two protein receptors on the platelets (P2Y1 and P2Y12), leading to ADP-mediated platelet aggregation with the glycoprotein IIb/IIIa (GPIIb/IIIa) complex pathway. As an ADP receptor inhibitor, clopidogrel prevents platelet aggregation, but it does not interfere with arachidonic acid metabolic pathways [8]. There are a lot of metabolic enzymes encoded by different genes that are involved in multistep clopidogrel biotransformations [9,10,11]. The absorption of clopidogrel from the digestive tract depends on p-glycoprotein intestinal epithelial cells expressed by the *ABCB1* gene [12]. The metabolism in the liver is mainly controlled by the cytochrome P450 2C19 gene (*CYP2C19*) (responsible for approximately 50% of active metabolites) and the cytochrome P450 3A4 gene (*CYP3A4*) (39.8% of active metabolites) [13]. The *P2RY12* gene encodes the P2Y12 receptor and *GPIIIa* gene–GPIIb/IIIa complex, a biologic activity target and effector of clopidogrel [14]. Genetic variations and metabolic disturbances at every single step of this complex process may influence the response to clopidogrel [15].

## 3. Platelet Function Testing

There are several platelet function assays that can be performed to assess the platelet reactivity. The most commonly used are turbidimetric aggerogometry (VerifyNow; Accumetrics Inc, San Diego, CA, USA), moderately high shear stress platelet function analysis (PFA-100; Siemens Medical Solutions USA, Inc, Malvern, PA, USA), light transmission aggregometry (LTA; Aggrecorder II; Menarini Diagnostics, Florence, Italy), vasodilator-stimulated phosphoprotein phosphorylation assay (VASP; Biocytex Inc, Marseille, France), and impedance aggregometry (Multiplate; Roche Diagnostics, Meylan, Rhone-Alpes, France). Most platelet function assays measure the level of platelet aggregation in response to a variety of agonists. A higher activation of platelets during clopidogrel therapy high on-treatment platelet reactivity is equivalent to a reduced effectiveness of a therapy. Different assays use specific cut-off points to identify clopidorel resistance or high on-treatment platelet reactivity, e.g., VerifyNow has more than 230 P2Y12 reactive units (PRU); Multiplate has more than 47 Units, PFA-100 has less than 106 s of closure time; LTA has an aggregation rate of >70% or >46%; VASP has a platelet reactivity index of greater than 50% [12,16,17]. Some authors have also used flow cytometry or thromboelastography (TEG) to assess platelet function in stroke patients [18,19]. Every platelet functon test has its own advantages and limitations. Nevertheless, none of them have a significant advantage over the others, which means that they all have equal importance and can be used interchangeably. Notably, weak or moderate but signifcant corrrelations between assays are usually observed, but some authors have demonstrated that LTA does nor correlate significantly with the others [16,20].

## 4. The Prevalence of Clopidogrel High On-Treatment Platelet Reactivity

The prevalence of clopidogrel resistance in ischemic stroke patients is variable and depends mainly on the method used for the assessment of platelet reactivity and the cut-off value used. In ischemic stroke research, the most frequently used technique to measure platelet aggregation is LTA. Prevalence rates fall into fairly wide ranges (16.83% to 50%) [12,16,17,18,21]. In point of care methods, the prevalence rates are as follows: VerifyNow—from 21.5% to 55.5% [16,22,23,24,25]; PFA—from 38.9% to 92% [16,25]; Multiplate—44% [26]; VASP—44.4% [16]; flow cytometry—from 32.8% to 56.1% depending on the selected surface antigens [18], and TEG—20.1% [19]. This wide range of results indicates that there is still a need to search for and improve methods for standardizing platelet function monitoring. Nevertheless, the above data suggest that clopidogrel resistance is as common as aspirin resistance [27] and may significantly reduce the effectiveness of the applied treatment in the secondary prevention of ischemic stroke.

## 5. Factors Influencing Clopidogrel Resistance

The response to the drug may be influenced by genetics or by clinical and pathophysiological factors (Figure 1).

### 5.1. Genetic Polymorphism

As mentioned above, many different genes are involved in the biotransformation of clopidogrel into the active substance. Their polymorphisms may interfere with the above process, leading to a lower amount of active metabolites, resulting in a reduced drug efficacy. The most common gene variant described in the literature related to clopidogrel resistance concerns *CYP2C19* polymorphisms. Recent studies have demonstrated that carriers with one or more *CYP2C19* loss-of-function alleles (2* or 3*) are at a higher risk of clopidogrel nonresponsiveness due to having lower levels of its active metabolite [9,14]. It is estimated that one-third of people have at least one loss-of-function allele (2* or 3*), and that percentage may be even greater among Asians [17]. Fu et al. revealed that among carriers with one or more *CYP2C19* loss-of-function allele, the percentage of individuals with clopidogrel resistance is estimated to be 71.7% which is significantly different from the prevalence of this phenomenon in noncarriers—32.1% [17]. Similarly, Jia et al. [28] demonstrated a higher incidence of clopidogrel high on-treatment platelet reactivity among ischemic stroke *CYP2C19* 2* or 3* carriers compared with noncarriers (58.5% vs. 37.4%; *p* = 0.02), and Liu et al. [9] showed that the estimated clopidogrel resistance risk of the *CYP2C19*2* allele carriers is higher than that of noncarriers (odds ratio (OR) = 2.366; *p* = 0.014). However, it is worth emphasizing that carriers of *CYP2C19* loss-of-function alleles were found to be significantly more likely to have clopidogrel high on-treatment platelet reactivity, but not an increased risk or higher recurrence of vascular events [17,29,30,31]. Another very important gene involved in the oxidation of clopidogrel in the liver is cytochrome *CYP3A4*. The *CYP3A4*1G* variant is considered to be a protective factor in clopidogrel resistance among ischemic stroke patients. It is estimated that the most frequent mutant allele is associated with greater liver metabolic activity, leading to a higher concentration of active clopidogrel-related substances, thus increasing platelet inhibition. Liu demonstrated that among *CYP3A4*1G* carriers, the estimated risk of clopidogrel resistance is significantly lower compared with that of noncarriers (OR = 2.360; *p* = 0.008) [9]. The *ABCB1* gene TT variant and the wild-type CC genotype have been associated with a higher risk of recurrent vascular events during clopidogrel treatment, but these findings were reported among coronary heart disease subjects [32,33]. In contrast, Su et al. [12] and Yi et al. [14] noticed no significant correlation between clopidogrel resistance and polymorphisms of the *ABCB1* gene in ischemic stroke subjects. However, the above studies were limited by a single locus assessment and small sample sizes. Polymorphisms of the *P2RY12* and *GPIIIa* genes have been shown to be associated with clopidogrel-dependent platelet reactivity [34,35,36]. Among stroke subjects, Yi et al. [14] showed that *P2RY12* TT+CT carriers are more likely to have a higher platelet reactivity following a clopidogrel treatment than *P2RY12* CC carriers, and Liu et al. [9] revealed that C allele carriers have lower rates of high on-treatment platelet reactivity compared with homozygotes (TT) (6.2% vs. 29.4%). Fu et al. [17] reported a trend toward a higher frequency of clopidogrel resistance in individuals with the *P2RY12* H2 haplotype compared with those with the H1 haplotype, but this result did not reach statistical significance. Yi et al. [14] found that *GPIIIa* AA carriers and AG+GG carriers differed significantly in terms of the incidence of platelet activity on clopidogrel therapy (*p* = 0.042). It has been noted that interactions between genes may have a greater impact on platelet activity than the effect of a single gene polymorphism. Moreover, it has been underlined that clopidogrel resistance may not be attributed to single genetic variants at individual loci alone but may be conditioned by multiple gene–gene intracations. Yi et al. [14] noted that the interactions of the *CYP2C19, P2RY12*, and *GPIIIa* variants are independently associated with the efficacy of clopidogrel. Furthermore, the synergistic effect of genes interactions may contribute not only to platelet activity but also to stroke prognosis. Yi et al. [37] showed that carriers of high-risk interactive genotypes not only exhibit a significantly lower platetet inhibition, but are also at a higher risk of early neurological deterioration than noncarriers (hazard ratio (HR) = 2.82; *p* = 0.003). The molecular mechanisms responsible for the above characteristics still remain unclear.

### 5.2. Concomitant Drugs

Most drugs are metabolized in the liver via similar enzyme pathways to the one that produces clopidogrel. Thus, the coadministration of several drugs may competitively reduce the effectiveness of each of them. The most commonly used concomitant drugs in the secondary prevention of stroke are the 3-hydroxy-3-methylglutaryl-coenzyme A reductase inhibitors (statins) and proton pump inhibitors, a protocol that is used to reduce the risk of developing gastrointestinal complications from antiplatelet agents. Most statins, including atorvastatin, lovastatin, or simvastatin, are extensively metabolized by the *CYP3A4* complex, while others, such as pravastatin or fluvastatin, do not. The related data on coronary heart disease are controversial. Some studies have not reported a negative interaction of statin coadministration (regardless of the pathway of biotransformation) with clopidogrel effectiveness in subjects undergoing coronary stenting [38]. In contrast, Lau et al. [39] demonstrated that the coadministration of atorvastatin competitively reduced platelet inhibition by clopidogrel proportionally to an increasing dose of statins. However, in ischemic stroke subjects, the coadministration of statins (regardless of the pathway of biotransformation) did not change the degree of platelet inhibition by clopidogrel [9,14,21,40]. However, Rath et al. [22] reported a higher percentage of statin coadministration in clopidogrel-resistant subjects compared with responders (25.4% vs. 12.3%), but the multivariate analysis did not show statistical significance (*p* = 0.25). To the best of our knowledge, there is currently no study indicating a significant negative impact of statin coadministration on the inhibitory effect of clopidogrel on platelet function in ischemic stroke patients.

Data on drug–drug interactions involving proton pump inhibitors and clopidogrel seem to be more conflicting. Several studies have shown that, due to the potent inhibition of *CYP2C19*, the coadministration of omeprazole or esomeprazole may be associated with a reduced efficacy of platelet inhibition by clopidogrel [41]. Moreover, the administration of clopidogrel in combination with omeprazole may be associated with a higher mortality rate and recurrent ischemic events [42]. Data on the above issues in patients with ischemic stroke are sparse and inconclusive. In a univariate analysis, Rath et al. [22] demonstrated an association between the coadministration of clopidogrel and a proton pump inhibitor (without specifying the type of drug) with high on-treatment platelet reactivity (*p* = 0.02), but these factors were not significantly correlated in a mutlivariate analysis (*p* = 0.10). Yi et al. [40] noted that clopidogrel resistance is not associated with the combined use of proton pump inhibitors (without specifying the type of drug) (*p* = 0.71). In contrast, Su et al. [12] demonstrated a highly significant association between the coadministation of proton pump inhibitors (not specified) and clopidogrel resistance in subjects vs. the achievement of a response (*p* < 0.001). Overall, based on the above inconsistent findings, proton pump inhibitors that are not involved in the *CYP2C19* pathway, such as pantoprazole, ranitidine, or famotidine, should be administered, if clinically warranted, in combination with clopidogrel to avoid competition in terms of the mechanism of action and to provide effective antiplatelet therapy [43].

### 5.3. Risk Factors for Vascular Disease Development

Many reports in the literature refer to the relationships of established risk factors for the development of vascular diseases, including stroke, with the antiplatelet effect of clopidogrel. Several studies have revealed that, in ischemic stroke subjects, clopidogrel high on-treatment platelet reactivity is significantly associated with diabetes [12,14,22,40]. Prolonged protein glycation as a result of hyperglycemia might reduce the biotransformation of clopidogrel and decrease the sensitivity of its receptors, leading to nonresponsiveness. Diabetes is currently considered to be the most important comorbid disease that can effectively limit the antiplatelet effect of the drug. Conversely, several authors did not not show similar findings regarding the prevalence of diabetes in stroke subjects [9,17]. This may suggest that the duration and the severity of diabetes play significant roles in modifying the antiplatelet effect.

Many interesting insights have resulted from studies on the influence of smoking on platelet reactivity during clopidogrel therapy. Many researchers have described the smokers’ paradox, which is that smokers experience better antiplatelet effects following clopidogrel treatment than nonsmokers, resulting in stronger platelet inhibition, in particular, in patients with coronary heart disease [44]. Kang et al. [45] and Rath et al. [22] showed that smokers who experience ischemic stroke exhibit lower PRU values, which is synonymous with a lower platelet reactivity, compared with nonsmokers. Similarly, Maruyama et al. [46] demonstrated that smokers achieve significantly lower platelet reactivity values (128 vs. 167 PRU, *p* = 0.002) than nonsmokers and, additionally, the incidence of clopidogrel resistance is significantly lower in smokers (12.9% vs. 25.9%; *p* = 0.033). Zhang et al. [47] concluded that the antiplatelet effect of clopidogrel in ischemic stroke patients is much better in smokers and showed a statistical trend for a reduced risk of recurrent vascular events with clopidorel than with aspirin [47]. The explanation of the above dependencies is believed to be the activation of the cytochrome P450 complex by nicotine, which may result in a more efficient metabolization of clopidogrel by the liver, leading to increased platelet inhibition [46]. Other risk factors for vascular diseases, such as hypertension, hiperlipidemia, atrial fibrillation, alcohol abuse, and obesity, do not contribute significantly to the modification of the antiplatelet effect of clopidogrel among ischemic stroke subjects.

### 5.4. Laboratory Findings

Jeon showed that clopidogrel resistance in ischemic stroke patients is significantly asssociated with a higher white blood count and C-reactive protein (CRP) level [48]. Similarly, Fu et al. noted a correlation between platelet reactivity and higher levels of high-specific C-reactive proteins [17]. Xanmemmedovr et al. [49] revealed that a higher platelet count (over 254 thousands/µL) is an independent factor that is associated with a higher platelet reactivity following treatment with clopidogrel. The remaining researchers did not emphasize the significant influence of laboratory tests, including markers of inflammation on the prothrombotic status, on the antiplatelet effect of clopidogrel, which suggests that this relationship is doubtful.

### 5.5. Etiology of Stroke

Previous studies have shown that aspirin resistance may be associated with a large vessel disease background in ischemic stroke patients [27]. Reports on the influence of the etiology of stroke on clodidogrel resistance are conflicting. Jeon showed that high on-treatment platelet reactivity occurs significantly more often among individuals with large vessel disease (a percentage of 80%) than in those with small vessel disease (6%) [48]. In contrast, Patel et al. [50] and Yi et al. [21] revealed that a lack of response to clopidogrel in large vessel disease patients is not more frequent than in small vessel disease ischemic stroke patients. Further research is needed to confirm the role of the etiology of stroke on clopidogrel resistance. For obvious reasons, patients with an embolic background of stroke, for whom anticoagulants are indicated, have not been analyzed.

## 6. Impact of Clopidogrel Resistance on Clinical Evaluation in Ischemic Stroke

### 6.1. Recurrent Vascular Events

The main goal of antiplatelet therapy as part of secondary prevention is to decrease the risk of recurrent ischemic events after the onset of ischemic stroke. The most common event is another recurrent stroke or transient ischemic attack, but stroke subjects are also at a higher risk of myocardial infarction (MI), sudden cardiac death, acute limb ischemia, or systemic embolism. Previous studies regarding aspirin resistance have demonstrated a significant, independent association between high on-treatment platelet reactivity and a higher risk of recurrent vascular events in stroke subjects [51]. The data on clopidogrel therapy are consistent. Yi et al. [40] showed that clopidogrel resistant subjects, followed for up to 6 months after stroke onset, experienced significantly more recurrent ischemic vascular events compared with clopidogrel responders (13.5% vs. 5.7%; *p* < 0.001). In particular, recurrent ischemic stroke was more strongly associated with the clopidogrel-resistant subgroup than the clopidogrel responders (10.5% vs. 3.5%; *p* < 0.001). However, the incidence of myocardial infarction and death due to vascular causes did not differ significantly between the above groups. Furthermore, in a multivariate regression analysis, the clopidogrel high on-treatment platelet reactivity was found to be the most significant independent risk factor for recurrent ischemic events (HR = 2.82; *p* < 0.01). In another study, also with a 6 month follow up period, in clopidogrel resistant subjects, a significant association with adverse clinical outcomes in all patients (*p* = 0.023) and in those with recurrent ischemic stroke (*p* = 0.021) was demonstrated, compared with clopidogrel responders [14]. Similar to the above, clopidogrel resistance was not shown to significantly increase the risk of myocardial infarction (*p* = 0.83) or sudden death of vascular origin (*p* = 0.82), but it was shown to be an independent predictor of adverse events (HR = 1.86; *p* = 0.042). A survival analysis by Fu et al. [17] conducted within 6 months of ischemic stroke indicated an increased risk of primary endpoints (composite of transient ischemic attack, ischemic stroke, myocardial infarction and sudden death) in subjects with clopidogrel resistance (20.6% vs. 7.3%, *p* = 0.04). Furthermore, clopidogrel high on-treatment platelet reactivity was considered an independent predictor of recurrent clinical events (HR = 3.1, *p* = 0.04). However, data on individual vascular events are lacking. Rao et al. [19] only followed up minor stroke subjects (National Institute of Health Stroke Scale (NIHSS) < 6 points) and high-risk TIA patients (ABCD2 scale > 3 points) for up to 90 days and used a tertile distribution map to assess platelet reactivity. The third tertile subgroup, which contained individuals with the highest values of platelet reactivity, had a higher rate of recurrent vascular events, including stroke, TIA, MI, lower extremity arterial disease, and all-cause deaths. A statistical analysis was not performed on individual events. In contrast, Yi et al. did not find any significant differences in the rates of recurrent nonfatal events (ischemic stroke *p* = 0.784, myocardial infarction *p* = 0.223; vascular death *p* = 0.752) between clopidogrel-resistant and sensitive subgroups [21]. However, the study was limited by a short follow up period—only 10 days after admission—which could undoubtedly have contributed to the different results obtained. In general, all of the above studies have similar limitations: a small sample size, use of a single platelet function test at a single time point, results with minor statistical significance, and a follow-up duration of less than six months. Nevertheless, despite these flaws and drawbacks, most of these studies confirmed that a good response to clopidogrel is a key element in the secondary prevention of recurrent vascular events, in particular, recurrent ischemic stroke.

### 6.2. Clinical Outcomes

Another very important issue related to the effective action of an antiplatelet agent is the reduction in the severity of ischemic stroke and the improvement of the prognosis and recovery. Therefore, it seems that clopidogrel resistance may significantly worsen the clinical and functional condition of stroke subjects and contribute to a poor prognosis and clinical outcome. Lee et al. [24] demonstrated that clopidogrel high on-treatment platelet reactivity is an independent risk factor associated with early neurological detorieration (defined as progression in the NIHSS by 1 point or more compared to the score at admission) in ischemic stroke patients (OR = 2.75; *p* = 0.029). However, only patients with large vessel disease as a cause of stroke were included in this study. Yi et al. [40] investigated the impact of clopidogrel resistance on late (at 6 month) functional outcomes, defined as favorable (modified Rankin Scale (mRS) 0–2 points) or unfavorable (mRS 3–6 points). In clopidogrel-sensitive subjects, a significantly higher percentage of individuals with a favorable prognosis was observed compared with clopidogrel nonresponders (79.9% vs. 67.5%; *p* = 0.001). At the same time, there was no significant difference in the clinical condition assessed with the NIHSS on the day of enrollment (acute phase of stroke) between clopidogrel responders and nonresponders (median 5.56 vs. 5.78; *p* = 0.16). Conversely, Jeon [48] showed that clopidogrel-resistant subjects had significantly severe initial clinical conditions (more points on the NIHSS) compared with sensitive subjects (median 10.7 vs. 6.0 *p* = 0.015). Qiu et al. [18] indicated that platelet reactivity had a predictive value for late prognosis (12 months after the onset of stroke). It was identified that a higher platelet reactivity following clopidogrel treatment occurred significantly more often among patients with poor outcomes (2 or more points on the mRS) (*p* = 0.001). Yi et al. [21] investigated the impact of clopidogrel resistance on early neurological deterioration (defined as an increase of 2 points or more within the 10 first days of onset on the NIHSS ). The frequency of early deterioration was significantly higher in the clopidogrel-resistant group compared with the clopidogrel responders (39.5% vs. 18%, *p* < 0.001). Subjects with clinical deterioration had significantly higher levels of platelet aggregation than patients without deterioration (*p* < 0.001). A regression analysis showed that clopidogrel resistance was an independent predictor of early neurological deterioration (HR = 2.76, *p* < 0.001). These data underscore the significant influence of adequate platelet inhibition by clopidogrel on the clinical course of stroke and both early and late prognoses.

### 6.3. Intracranial Bleeding

Intracranial hemorhage, such as gastrointestinal or urinary bleeding, may be one of the adverse effects of antiplatelet therapy that could worsen the clinical condition and recovery. Some researchers have investigated whether the degree of platelet inhibition could contribute to symptomatic bleeding. Lee et al. [24] showed that an early hemorrhagic transformation of ischemic stroke (assessed on the fifth day after stroke onset by magnetic resonanse imaging) is not associated with clopidogrel resistance (OR = 0.60; *p* = 0.484). Similarly, Yi et al. [21] reported that an early hemorrhagic transformation of ischemic stroke or early intracranial hemorrhage (up to 10 days after the onset of stroke) is not related to the incidence of clopidogrel high on-treatment platelet reactivity (respectively, *p* = 0.992; *p* = 0.753). In another study, regarding late complications, Yi et al. [40] followed up stroke subjects for 6 months and demonstrated no significant differences between clopidogrel-resistant and sensitive groups in terms of the rates of asymptomatic hemorrhagic transformation (*p* = 0.95) and intracranial hemorrhage (*p* = 0.92). The above results suggest that the sensitivity of platelets to clopidogrel is not a risk factor for either early or late intracranial bleeding.

## 7. Recommendations

Due to the lack of support by large, randomized, multicenter, clinical trials, the assessment of platelet reactivity in ischemic stroke patients is not currently recommended for routine use. However, numerous significant relationships concerning multiple effects on the course of stroke suggest that, in some clinical situations, the determination of the degree of platelet inhibition may be useful for modifying or intensifying the efficacy of treatment. Based on the provided data, this group should especially include subjects with recurrent ischemic stroke or other vascular events, nonsmokers with long-term and advanced diabetes, and carriers of genetic variants promoting clopidogrel resistance, if available. Platelet function monitoring and confirmation of high platelet activity should be a sufficient reason to switch the antiplatetet agent to improve the effectiveness of secondary prevention treatment. Reports on therapy modification based on platelet function testing are negligible and inconclusive. In a retrospective study, Depta et al. [52] showed that stroke patients did not benefit from drug switching based on platelet function. Furthermore, the modification of antiaggregative therapy could contribute to a higher risk of sudden death, bleeding, or recurrent vascular events. In contrast, Yi et al. [53], in the retrospective but multicenter study, demonstrated that the platelet function-guided modification of antiplatelet treatment can benefit stroke subjects by reducing the risk of recurrence of early ischemic events (HR = 0.67, *p* = 0.01), while not increasing the risk of sudden death or bleeding. Unfortunately, both studies included subjects based only on high on-treatment platelet reactivity, without differentiating them by additional determining factors which, in view of the above considerations, constitutes a limitation. Additionally, it should be emphasized that, compared to coronary artery disease, the capacity to modify antiplatelet treatment in stroke is significantly limited, as current guidelines recommend, apart from clopidogrel, only aspirin or dipyridamole.

## 8. Materials and Methods

This comprehensive review included papers regarding connections between clopidogrel and platelet reactivity, in particular, among ischemic stroke subjects. Clinical databases were searched by two independent researchers. English studies of all types, especially clinical trials, orginal research, observational, cross-sectional and cohort studies, and reviews were included. The various stages of proper selection of data from the literature are presented in Figure 2.

## 9. Conclusions

In summary, clopidogrel resistance in ischemic stroke patients was presented as a common and important phenomenon with a multifactorial background. Similar to aspirin resistance, it has a negative impact on the clinical course of stroke, worsens the early- and long-term prognoses and increases the risk of recurrent vascular events. This highlights the importance of effective antiplatelet therapy in preventing the occurrence of stroke and the growing role of platelet function monitoring to identify nonresponders and support personalized and improved treatments. Specific features of ischemic stroke patients who are particularly at risk of developing clopidogrel resistance have been identified, and these should be considered and used as indicators of platelet reactivity testing to ensure the maximum efficacy of antiplatelet therapy.

## Figures and Tables

**Figure 1 ijms-21-06408-f001:**
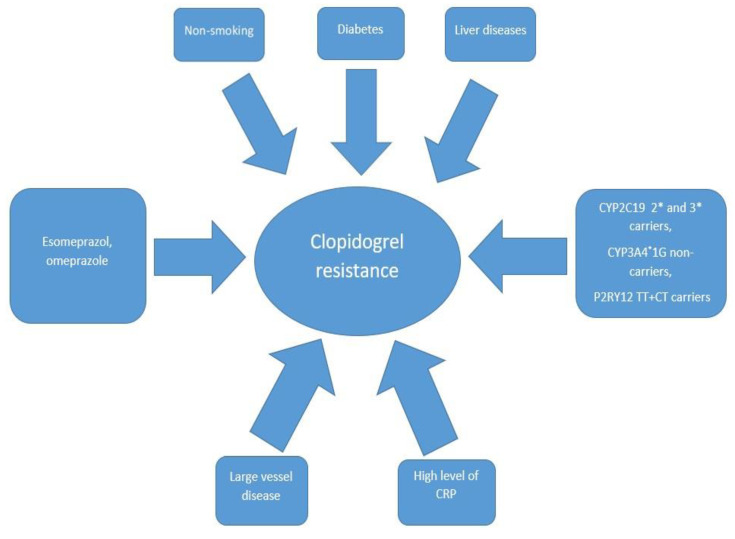
The most common predictors associated with clopidogrel resistance. CRP—C-reactive protein, CYP2C19 2* and 3*—cytochrome P450 2C19 gene with loss-of-function 2* and 3* alleles, CYP3A4*1G- *—cytochrome P450 3A4 gene with loss-of-function 1G allele, P2RY12 TT+CT—P2Y12 receptor gene with double T alleles or C and T alleles.

**Figure 2 ijms-21-06408-f002:**
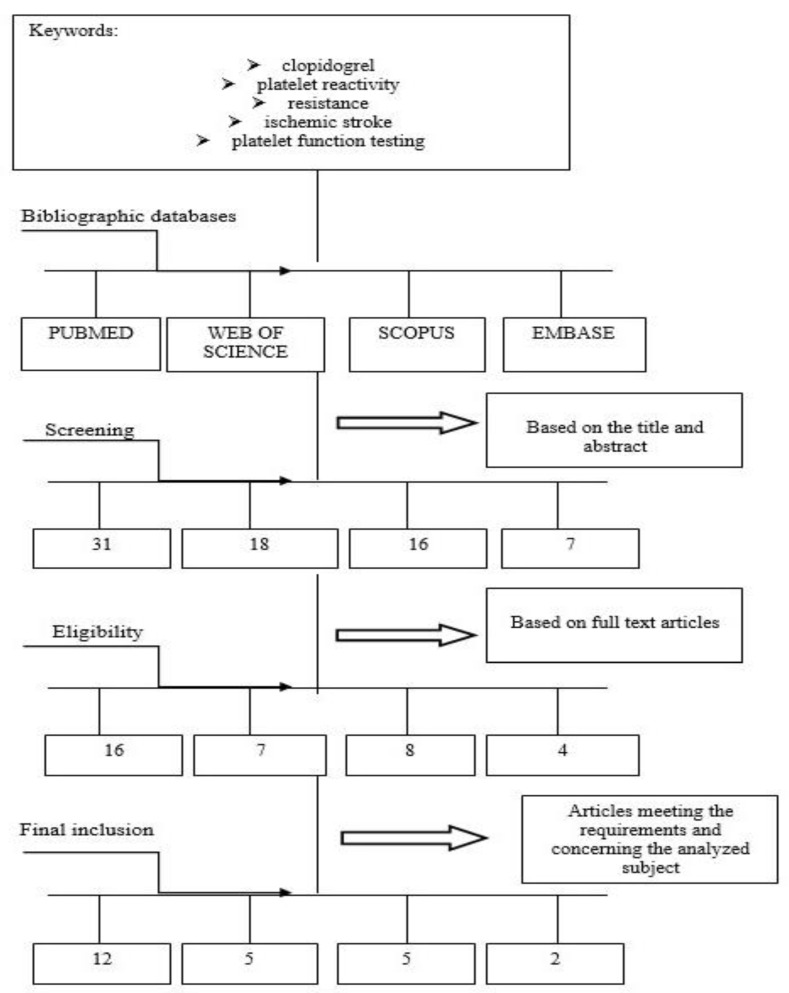
Diagram showing the process leading to the selection of the appropriate articles.

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
