# Peer review of "The Phenomenon of Clopidogrel High On-Treatment Platelet Reactivity in Ischemic Stroke Subjects: A Comprehensive Review"

_ijms, 2020, doi:10.3390/ijms21176408_

Round 1
Reviewer 1 Report
This is a review of the role of clopidogrel high on-treatment platelet reactivity (HTPR) in ischemic stroke patients. It is a very well organized, useful, and informative review. However, the relative studies (reviews) of the role of clopidogrel high on-treatment platelet reactivity in ischemic stroke were reviewed before. It's unclear if this review has novelty to attract the reader's attention.
Line 58, “Molecular mechanism of action” should be “Molecular mechanism of action of clopidogrel”
The authors did a sloppy job on forming Figures. Figures 1 and 2 have some flaws (red squiggly lines). Please redo them.
Author Response
Response to Reviewer 1 Comments
At the beginning, I would like to thank You for the careful review of our study and the constructive comments that have been used to organize all issues and improve the work.
Point 1:
This is a review of the role of clopidogrel high on-treatment platelet reactivity (HTPR) in ischemic stroke patients. It is a very well organized, useful, and informative review.
Response:
We are grateful to the Reviewer for the important opinion about useful and informative data reported in our paper. We have made every effort to prepare the manuscript reporting the role of clopidogrel resistance in ischemic stroke subjects. We are satisfied that our hard work has been appreciated.
Point 2:
However, the relative studies (reviews) of the role of clopidogrel high on-treatment platelet reactivity in ischemic stroke were reviewed before. It's unclear if this review has novelty to attract the reader's attention.
Response:
To the best of our knowledge there has been no comprehensive review of this topic. Most of the similar papers were about aspirin resistance or were too general - they didn't focus on ischemic stroke patients. Hence, in our opinion, there was a need to create this manuscript containing current, complex and relevant information on this subject.
Point 3:
Line 58, “Molecular mechanism of action” should be “Molecular mechanism of action of clopidogrel”.
Response:
We appreciate this comment. We have corrected it as suggested by Reviewer.
Point 4:
The authors did a sloppy job on forming Figures. Figures 1 and 2 have some flaws (red squiggly lines). Please redo them.
Response:
Thank You for this suggestion. According to the Reviewer’s recommendation we have improved our figures, corrected flaws and drawbacks to make them more clear and transparent.
Thank You very much for all criticism. We hope that revision of our paper made according the Reviewer’s guidelines will improve our manuscript.
Reviewer 2 Report
The manuscript will benefit from some revision from clarity and style.
In particular, the title is not clear and the aim of the review in the abstract is not clearly stated (while, for example, it is much clearer in the introduction).
The expression “High on-treatment platelet reactivity” should be explained in the abstract and possibly introduction as it is hard to understand for readers who are not familiar with the term.
I would not call clopidogrel as an inhibitor but an antagonist.
Figure 1: maybe liver diseases could be added to the figure.
Author Response
Response to Reviewer 2 Comments
At the beginning, I would like to thank You for the careful review of our study and the constructive comments that have been used to organize all issues and improve the work.
Point 1:
The manuscript will benefit from some revision from clarity and style.
Response:
We are grateful to the Reviewer for this important remark. We have made every effort to revise the manuscript to make it more transparent. As suggested by the Reviewer we thoroughly reviewed our study and improved the speech, style and lexical accuracy. We modified the manuscript to make it more comprehensive.
Additionally, according the Reviewer’s recommendations we have made extensive changes of language and structure in our manuscript, using the MPDI professional English editing service.
Point 2:
In particular, the title is not clear and the aim of the review in the abstract is not clearly stated (while, for example, it is much clearer in the introduction).
Response:
Thank You for this suggestion. We have corrected the abstract section to underline the aim of the review (Lines 17-20).
As You suggested- we modified also the title of the manuscript to improve the clearance.
Point 3:
The expression “High on-treatment platelet reactivity” should be explained in the abstract and possibly introduction as it is hard to understand for readers who are not familiar with the term.
Response:
We fully agree with the Reviewer opinion. According to the Reviewer’s recommendation we have extended the abstract sections to explain this expression (Lines 14-16). It seems that this procedure will make it easier for readers unfamiliar with the topic to introduce it better. It is also explained in the main text- Lines 88-89.
Point 4:
I would not call clopidogrel as an inhibitor but an antagonist.
Response:
We appreciate this remark. We have corrected it (Line 35).
Point 5:
Figure 1: maybe liver diseases could be added to the figure.
Response:
According to the Reviewer’s suggestion we have added liver diseases to our Figure 1.
Thank You very much for all criticism.
We hope that revision of our paper made according the Reviewer’s guidelines
will improve our manuscript.